

# Phytochemical production and antioxidant activity improvement of *Rhinacanthus nasutus* (L.) Kurz calli by *in vitro* polyploidization

Wipa Yaowachai[1], Prathan Luecha[2] and Worasitikulya Taratima[1]

[1] Department of Biology, Faculty of Science, Khon Kaen University, Khon Kaen, Thailand
[2] Department of Pharmacognosy and Toxicology, Faculty of Pharmaceutical Science, Khon Kaen University, Khon Kaen, Thailand

## ABSTRACT

**Background:** *Rhinacanthus nasutus* (L.) Kurz is a multipurpose ethnomedicinal shrub containing various bioactive compounds and phytochemicals. Inducing polyploidy is an alternative way to enhance the production of secondary metabolites in medicinal plants. The main objectives of this research were to study the effect of polyploidization on the phytochemical content and antioxidant activity of *R. nasutus* calli.

**Methods:** *In vitro* polyploidy was induced by soaking calli in colchicine at different concentrations and for different exposure times. To determine callus polyploidy, the relative DNA contents of each sample were examined using flow cytometry. Diploid, tetraploid, and mixoploid calli were extracted to determine the total phenolic content (TPC), total flavonoid content (TFC), and antioxidant activity.

**Results and conclusion:** Results showed that the callus survival rate decreased with increasing colchicine concentration and exposure time. The highest percentage of induced tetraploid (66.67%) and mixoploid (66.67%) calli were obtained at 0.05% and 0.2% colchicine with exposure times of 48 and 24 h, respectively. Tetraploid calli showed the highest TPC (81.28 mg GAE/g extract), TFC (35.33 mg QE/g extract), and antioxidant activity compared to diploid and mixoploid calli. Additionally, tetraploid calli demonstrated an approximately twofold greater increase in TPC and TFC compared to diploid calli. The analysis of polyploid callus samples revealed that tetraploid calli exhibited the highest antioxidant activity, whereas diploid calli demonstrated the lowest antioxidant activity across all applied assays. Therefore, inducing a tetraploid of *R. nasutus* calli plays a critical role in modifying phytochemical content and antioxidant activity.

# INTRODUCTION

*Rhinacanthus nasutus* is a medicinal herb that belongs to the Acanthaceae family, also known as snake jasmine or Thong Pan Chang (in Thai). *R. nasutus* has been identified as a valuable medicinal plant because of its therapeutic effects for a variety of diseases. The

Corresponding author
Worasitikulya Taratima,
worasitikulya@gmail.com

seeds, roots, and leaves have traditionally been used to treat scabies, eczema, and other skin diseases (*Brimson et al., 2020*). Several phytochemicals including triterpene derivatives, O-methylated flavone, plant sterols, 7-hydroxycoumarin, anthraquinone, benzoate ester, organic compounds, rhinacanthin-A, -B, -C, and -D and quinol have been isolated from the roots, stems, and leaves of *R. nasutus* (*Tewtrakul, Tansakul & Panichayupakaranant, 2009*). The major bioactive compounds are rhinacanthin-C, -D, and -N. Rhinacanthin-C has protective effects against diabetic nephropathy, attributable to the reduction of oxidative stress and inflammation in the nephrons (*Zhao et al., 2019*). Consequently, *R. nasutus* is a valuable therapeutic herb in terms of the quantity and quality of bioactive compounds.

The most common way to propagate *R. nasutus* is by stem cuttings and seeds, which fall readily and are weather-sensitive. *In vitro* propagation is another alternative method to rapidly multiply large numbers of plants and secondary metabolites. Nowadays, *in vitro* culture systems (callus culture, cell suspension culture, and organ culture) are regarded as crucial biotechnological instruments and sources of secondary metabolite synthesis (*Efferth, 2019*). As previously reported, calli derived from nodes of *R. nasutus* showed higher phenolic content and antioxidant activity than young leaf and mature leaf explants (*Yaowachai, Luecha & Taratima, 2023*). Several studies have investigated techniques for enhancing the production of secondary metabolites in medicinal plants. Artificial polyploidization is one strategy that has been successfully utilized to increase biomass and secondary metabolite content in several ornamental and medicinal plant species (*Mohammadi et al., 2023*).

Polyploidy is typically defined as a cell or organism having more than two complete sets of chromosomes. *In vitro* polyploidization relies on an interrupting mechanism in which the cell suppresses interpolar microtubules during anaphase. Chromosome doubling agents such as colchicine, oryzalin, and trifluralin are commonly used to induce polyploidization. Colchicine, an alkaloid substance found in *Colchicum autumnale* bulbs, plays an important role in degrading microtubules and inhibiting tubulin polymerization (*Akram et al., 2012*). Colchicine disrupts cell division by attaching to β-tubulin and forming a unit which destabilizes microtubules resulting in microtubule breakdown (*Leung, Hui & Kraus, 2015*).

Colchicine has been effectively utilized for genome doubling in a wide range of species. *e.g.*, *Nicotiana tabacum* (*Sood, Dwivedi & Reddy, 2013*), *Zea mays* (*Melchinger et al., 2016*), *Tagetes erecta* (*He et al., 2016*), *Brassica oleracea* var. acephala (*Chen et al., 2022*), *Centella asiatica* (*Surson, Sitthaphanit & Wongkerson, 2024*), *Lycium chinense* (*Zhang et al., 2024*). In plants, colchicine is commonly utilized to disrupt chromosome separation at metaphase, facilitating the induction of polyploidy (*Bhattacharyya et al., 2008*). Some polyploids are known for growing larger organs for agronomic purposes, with altered secondary metabolite production for medicinal plants. This phenomenon is known as the Gigas effect, in which cell size increases because of chromosomal doubling (*Chaves et al., 2018*). The Gigas effect, establishing polyploidy as a critical approach for optimizing crop yields and enhancing the pharmaceutical properties of medicinal plants.

Ploidy level can be assessed through direct methods, such as chromosome counting, or indirectly using techniques like flow cytometry, stomatal size measurement, guard cell chloroplast counting, and morphological analysis. Historically, determining ploidy levels in plants relied exclusively on karyotypic evaluations involving metaphase chromosome counting. However, advancements in techniques, particularly flow cytometry, have revolutionized ploidy analysis (*Ochatt, Patat-Ochatt & Moessner, 2011*). This method offers significant advantages, including high throughput, accuracy, resolution, minimal plant damage, and cost-effectiveness per sample, making it a superior alternative to traditional methods for assessing ploidy and genome size (*Sliwinska, 2018*).

The chromosome number of *R. nasutus* has not been extensively documented in scientific literature. However, cytological studies on *R. nasutus* have previously reported chromosome counts of *n* = 15 (*Saggoo, 1983*) and 2*n* = 30 (*De, 1966*; *Löve, 1976*; *Kumar & Subramaniam, 1986*). Therefore, plant tissue culture techniques and polyploidy induction serve important functions in micropropagation of *R. nasutus* and secondary metabolite production. This study elucidated the increase of secondary metabolite production and antioxidant activity in the colchicine-based protocol for polyploidy induction of *R. nasutus* callus.

## MATERIALS AND METHODS

### *In vitro* callus induction

*R. nasutus* calli were successfully induced from node explants developed in our previous study (*Yaowachai, Luecha & Taratima, 2023*). Initially, the nodes were collected and washed under running tap water for 15 min. The cleaned nodes were then immersed in 70% ethanol for 1 min, followed by aseptic disinfection for 15 min using a 0.1% (w/v) solution of mercuric chloride ($HgCl_2$). Subsequently, the nodes were rinsed three times with sterile distilled water for 5 min. Finally, the nodes were placed on MS medium containing 30 g/L sucrose to induce shoot growth. *In vitro*-derived nodes of *R. nasutus* were used to generate the calli because their total phenolic content and antioxidant activity were higher than in young and mature leaf explants. In our study, *in vitro*-derived nodes were grown from the same accession in order to obtain samples with the same genotype. The sterile nodes were cut into 0.5 cm lengths and transferred to *Murashige & Skoog (1962)* solid medium supplement with 1 mg/L kinetin and 1 mg/L 2,4-D. All media were solidified using 8 g/L of agar, with pH adjusted to 5.8 before autoclaving. The calli were cultured at 16/8 h light/dark cycle under white cool fluorescent light at 1,800 lux and 25 ± 2 °C for 4 weeks.

### *In vitro* polyploidy induction

Four-week-old *R. nasutus* calli were used to induce polyploidy. Colchicine (LobaChemie Pvt. Ltd., Mumbai, India) was prepared in sterilized distilled water and then filtered through a sterile filter membrane with a pore size of 0.22 μm. The colchicine concentrations used in this study were selected based on previous literature reports that investigated the effects of colchicine on polyploidization in various plant species with slight modifications (*Sadat Noori et al., 2017*). The calli were soaked in a filter-sterilized

colchicine solution with concentrations of 0%, 0.05%, 0.1%, and 0.2%, while being continuously shaken on a shaker at 120 rpm for 24 and 48 h. It must be kept away from light during the curing process. The soaked calli were rinsed three times with sterilized distilled water and transferred to culture media. The MS medium supplemented with 1 mg/L kinetin and 1 mg/L 2,4-D was used for culturing calli. All treated calli were cultured at 25 ± 2 °C with 16/8 h (light/dark) photoperiod under white cool fluorescent light at 1,800 lux for 4 weeks.

## Flow cytometry analysis

The nuclear DNA content in the colchicine-treated and control calli was determined by flow cytometry analysis as described by Quantum Analysis GmbH, Germany. The whole living part of both treated and control (0.5 g) calli were collected and chopped with a sharp new blade in 500 µL of Quantum Stain NA UV 2 (A) as the buffer solution. Then, polyvinyl-pyrrolidone (PVP) was added to clear the debris. The nuclei suspension was filtered through a nylon filter with a 30 µL mesh diameter into a tube followed by staining the nuclei with 500 µL of Quantum Stain NA UV 2 (B). After 1 min, the nuclear suspensions were analyzed with a flow cytometer (Quantum P Flow Cytometer; Quantum Analysis, Münster, Germany) and expressed as flow cytometry histograms showing the quantitative measurements of the nuclear DNA content in c-units (x-axis) and number of cells (y-axis). The ploidy levels of the calli were identified as diploid, tetraploid or mixoploid.

## Preparation of *R. nasutus* callus extracts

Calli identified as diploid, tetraploid, and mixoploid were used for extraction to analyze the phytochemical content and antioxidant activity as described by *Yaowachai, Luecha & Taratima (2023)*. The samples were hot air oven-dried at 50 °C for 7 days and ground into a fine powder using a grinder. The powdered materials (1 g) were extracted with methanol (10 mL) by ultrasound-assisted extraction (UAE) at 40 kHz, 30 °C for 30 min. The extraction procedures were repeated nine times. The sample extracts were combined, filtered with filter paper (11 µm pore size) and evaporated to dryness in SpeedVac equipment at 45 °C for 16 h. The extracts were kept at −20 °C for further analysis. Percentage of extraction yield was calculated using the following formula:

$$\text{Percentage of extraction yield} = (\text{actual yield}/\text{theoretical yield}) \times 100 \qquad (1)$$

## Determination of total phenolic content

The total phenolic content (TPC) of *R. nasutus* calli (diploid, tetraploid, and mixoploid) was determined using the Folin-Ciocalteu method, as described by *Hmamou et al. (2022)* with slight modifications. Briefly, 20 µL of each crude extract (1 mg/mL in methanol) was mixed with 100 µL of 10% (v/v) Folin-Ciocalteu reagent and 80 µL of 7% (w/v) sodium carbonate ($Na_2CO_3$) in a 96-well microplate. The mixture was incubated in the dark for 30 min at room temperature, and the absorbance was measured at 760 nm using a microplate spectrophotometer and each procedure was repeated five times. A standard was

prepared using 2-fold dilutions of gallic acid (5–320 µg/mL) and calculated using the linear equation from the calibration curve (y = 0.0043x + 0.0589, $R^2$ = 0.9994). The total phenolic content (TPC) was expressed as milligrams of gallic acid equivalent per gram dry weight of crude extract (mg GAE/g extract).

## Determination of total flavonoid content

The total flavonoid content (TFC) of *R. nasutus* calli (diploid, tetraploid, and mixoploid) was determined using the aluminum chloride method as described by *Kim, Kim & Shin (2024)* with slight modifications. Briefly, a 10 µL sample (1 mg/mL in methanol), was mixed with 6 µL of 5% (w/v) sodium nitrite ($NaNO_2$) for 5 min, then 12 µL of 10% (w/v) aluminum chloride ($AlCl_3$) was added. The mixture was mixed and incubated in the dark for 5 min. Finally, 60 µL of 1 M sodium hydroxide (NaOH) was added, followed by 112 µL of deionized water to make up the volume to 200 µL. The absorbance was measured at 510 nm using a microplate spectrophotometer and each procedure was repeated five times. A standard solution was prepared using quercetin (5–320 µg/mL) and calculated using the linear equation from the calibration curve (y = 0.0003x + 0.0012, $R^2$ = 0.9992). The total flavonoid content was expressed as milligrams of quercetin equivalent per gram dry weight of crude extract (mg QE/g extract).

## Evaluation of antioxidant activities by FRAP assay

The ferric reducing antioxidant potential (FRAP) assay of *R. nasutus* calli (diploid, tetraploid, and mixoploid) was conducted according to *Niroula et al. (2021)* with slight modifications. The FRAP reagent was prepared by mixing acetate buffer (300 mM, pH 3.6), 2,4,6-Tris(2-pyridyl)-s-triazine (TPTZ; 10 mM in 40 mM of HCl), and $FeCl_3$ (20 mM in distilled water) in a ratio of 10:1:1 (v/v/v) and incubated for 10 min before use. Briefly, 10 µL of each extract (2 mg/mL) was mixed with 190 µL of FRAP reagent and incubated in dark at 37 °C for 15 min. The absorbance was measured at 593 nm using a microplate spectrophotometer and each procedure was repeated five times. A calibration curve with Trolox standards was measured in the range 2.50–320 µg/mL (y = 0.0041x + 0.044, $R^2$ = 0.9996). All values were expressed as mg Trolox equivalent per g dry weight extract (mg TE/g extract).

## Evaluation of antioxidant activities by DPPH assay

The free radical scavenging activity of each extract was determined using the 2,2-diphenyl-1-picrylhydrazyl (DPPH) assay as described by *Li et al. (2021)* with slight modifications. Briefly, 100 µL of serially diluted samples with methanol were mixed with 100 µl of DPPH solution (dissolved in methanol) and incubated in darkness at room temperature for 30 min. The absorbance of the reaction mixture was measured at 517 nm using a microplate spectrophotometer and each sample was repeated five times. Ascorbic acid was used as a standard and methanol served as the negative control. The DPPH radical scavenging activity of extracts and standard as percentage of free radical scavenging activity (FRSA) was calculated using the following formula:

$$\%FRSA = [1 - (A\ sample/A\ control)] \times 100 \qquad (2)$$

where A sample is the absorbance of the sample solution, and A control is the absorbance of negative control. The result of each extract was expressed as the half-maximal inhibitory concentration ($IC_{50}$). The $IC_{50}$ value was determined by plotting the percentage of free radical scavenging activity against the concentration of each extract.

## Evaluation of antioxidant activities by ABTS assay

The 2,2-azino-bis-3-ethylbenzothiazoline-6-sulfonic acid (ABTS) radical scavenging assay was determined as described by *Ehiobu, Idamokoro & Afolayan (2021)* with slight modifications. $ABTS^{\bullet+}$ radical cation was prepared by mixing 7.0 mM ABTS solution and 2.45 mM potassium persulfate 1:1 (v/v) and incubated in darkness at room temperature for 12–16 h. Then, the $ABTS^{\bullet+}$ solution was diluted with distilled water to obtain absorbance at 734 nm between 0.7 ± 0.03. For the analysis, 50 µl of each diluted sample was mixed with 150 µl of $ABTS^{\bullet+}$ solution and incubated in the darkness for 10 min. The absorbance of the reaction mixture was measured at 734 nm using a microplate spectrophotometer and each sample was repeated five times. The percentage of free radical scavenging activity was calculated as described in Eq. (2) and the result was expressed as the $IC_{50}$ value using ascorbic acid as a standard.

## Statistical analysis

All experiments were set up as completely randomized designs. Statistical analysis was conducted using SPSS software (IBM SPSS® Statistics version 22.0). In this study, the number of replicates per treatment varied according to the measured parameter: survival rate (12 replicates); fresh weight, dry weight, DPPH, and ABTS assays (five replicates each); polyploidy induction rate and extraction yield (three replicates each); and TPC and TFC (10 replicates each). The data were analyzed by one-way analysis of variance (ANOVA) and expressed as mean values ± standard error (SE). Duncan's multiple range test was used for the comparison of means between groups with 95% significance ($p < 0.05$). Correlations were conducted using OriginPro® 2024 (OriginLab, Northampton, MA, USA). Principal component analysis (PCA) was performed using R Studio software (Posit Software, PBC). Correlations were presented as values of Pearson's correlation coefficient, with significant differences in correlations determined at the $p < 0.05$ level. PCA was used to analyze interrelationships between the variables by Pearson-type matrices.

# RESULTS

## *In vitro* polyploidy induction

The survival rates of calli treated with colchicine were used to determine the optimal colchicine dose and exposure time. Various concentrations of colchicine for two exposure times gave different callus survival rates (Table 1). The highest survival rate was observed in the media without colchicine at 24 and 48 h exposure times. Calli treated at the highest colchicine concentration with two exposure times had reduced survival rate. The negative relationship between survival rate and colchicine concentration was predictable in this study.

**Table 1 Effects of different colchicine concentrations and exposure times on polyploidy induction of _R. nusutus_ calli.**

| Colchicine concentrations (% w/v) | Exposure times (h) | Survival rate (%) | Polyploidy induction by ploidy level (%) | | |
|---|---|---|---|---|---|
| | | | Diploid | Tetraploid | Mixoploid |
| 0 | 24 | 100.00 ± 0.00[a] | 100.00 ± 0.00[a] | 0.00 ± 0.00[b] | 0.00 ± 0.00[b] |
| | 48 | 100.00 ± 0.00[a] | 100.00 ± 0.00[a] | 0.00 ± 0.00[b] | 0.00 ± 0.00[b] |
| 0.05 | 24 | 75.00 ± 13.06[ab] | 100.00 ± 0.00[a] | 0.00 ± 0.00[b] | 0.00 ± 0.00[b] |
| | 48 | 75.00 ± 13.06[ab] | 33.33 ± 19.25[b] | 66.67 ± 33.33[a] | 0.00 ± 0.00[b] |
| 0.1 | 24 | 75.00 ± 13.06[ab] | 66.67 ± 33.33[ab] | 33.33 ± 19.25[ab] | 0.00 ± 0.00[b] |
| | 48 | 58.33 ± 14.86[b] | 66.67 ± 33.33[ab] | 33.33 ± 19.25[ab] | 0.00 ± 0.00[b] |
| 0.2 | 24 | 41.67 ± 14.86[b] | 33.33 ± 19.25[b] | 0.00 ± 0.00[b] | 66.67 ± 33.33[a] |
| | 48 | 41.67 ±14.86[b] | 100.00 ± 0.00[a] | 0.00 ± 0.00[b] | 0.00 ± 0.00[b] |

**Note:**

Mean ± SE values followed by different superscripts in the same column are significantly different according to ANOVA and Duncan's Multiple Range Test ($p < 0.05$).

Colchicine-treated calli were identified as diploid, tetraploid, and mixoploid by flow cytometry analysis. For tetraploid calli, the highest polyploidy induction rate was recorded in 0.05% colchicine solution at 48 h (66.67%), followed by 0.1% colchicine solution at 24 and 48 h (33.33%) (Table 1). Mixoploid calli were only observed in 0.2% colchicine solution at 24 h (66.67% polyploidy induction). From flow cytometry analysis, the ploidy level of the surviving calli was determined by a fluorescence intensity histogram. The fluorescence intensity peak positions of diploid (2C) and tetraploid (4C) were recorded at channels 200 and 400, respectively (Figs. 1A, 1B). The peaks of mixoploid (2C + 4C) calli were expected at both channels 200 and 400 (Fig. 1C). The histogram also showed that tetraploid calli successfully duplicated the 2C DNA by doubling the amount of DNA content. The histogram peaks were calculated by the relative fluorescence of DNA in the G1 phase, which C-value is the amount of DNA in a haploid genome (_Greilhuber et al., 2005_). In the G1 phase, somatic cells have a DNA content of 2C, which increases to 4C in the G2 phase. Therefore, diploid calli contain a small peak of G2 phase nuclei at channel 400, which has a DNA content of 4C (Fig. 1A). Similarly, tetraploid calli contains a small peak of G2 nuclei at channel 800, which has a DNA content of 8C (Fig. 1B). A calli with approximately equal amounts of 2C and 4C is identified as a mixoploid calli (Fig. 1C).

Diploid, tetraploid, and mixoploid calli had fresh weights of 3.04, 3.45, and 2.72 g, respectively. These were not statistically significantly different ($p > 0.05$), as shown in Table 2. The diploid calli had a higher dry weight (0.15 g) than the tetraploid (0.14 g) and mixoploid calli (0.11 g), as shown in Table 2. Surviving colchicine-treated and untreated calli showed pigments of light green to dark green, while dead explants turned a dark brown color. Stereomicroscopic observation showed that callus tissues were compact type (Fig. 2).

### Phytochemical analysis

The diploid, tetraploid, and mixoploid calli were extracted by UAE using methanol, with percentage extraction yields shown in Table 2. The tetraploid calli had significantly ($P$-value = 0.00) highest extraction yield of 27.56%, followed by mixoploid and diploid calli

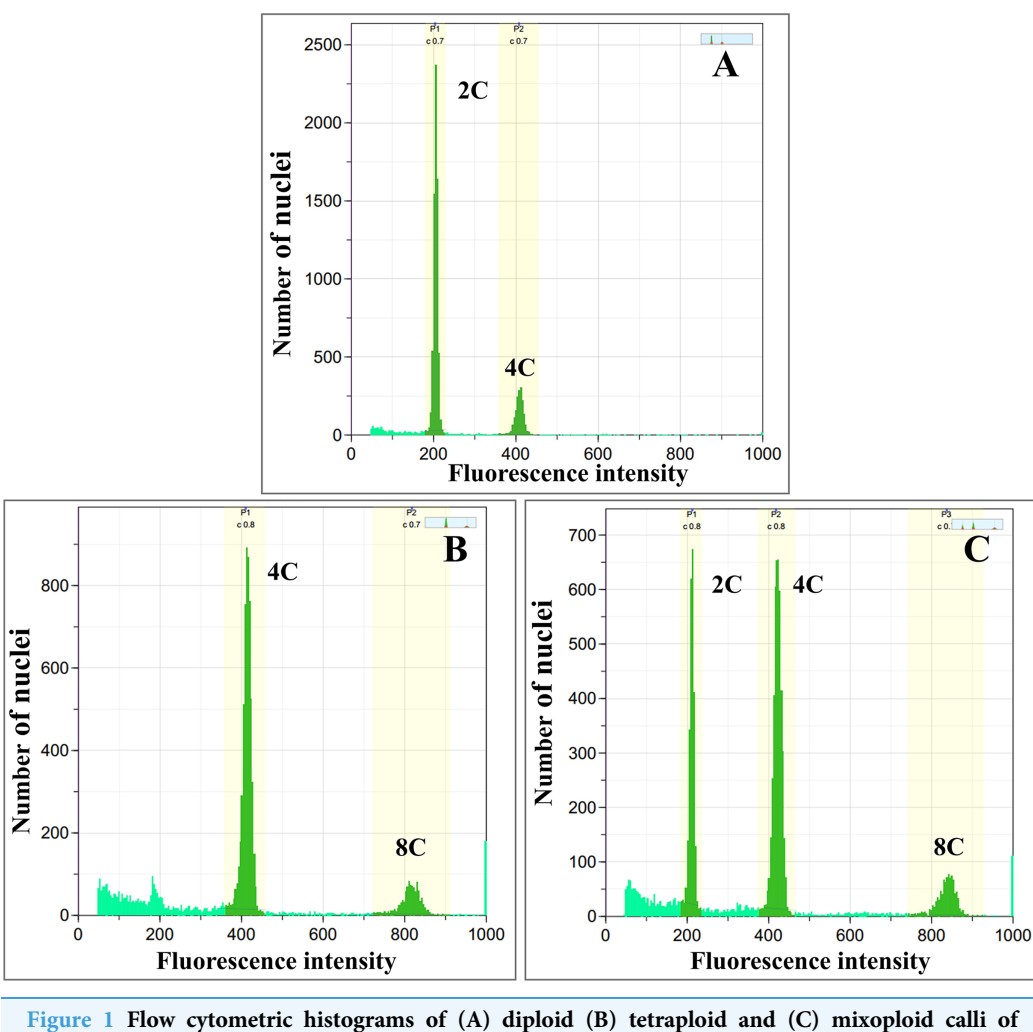

**Figure 1 Flow cytometric histograms of (A) diploid (B) tetraploid and (C) mixoploid calli of *R. nasutus*.** C-value: the amount of DNA in a haploid genome.

at 14.14% and 11.28%, respectively. The effective influence of polyploidy induction on the optimal production of phytochemical contents in *R. nasutus* calli was investigated. Total phenolic content values obtained from the diploid, tetraploid, and mixoploid callus extracts were analyzed using the Folin-Ciocalteu method, with results shown in Table 2. The tetraploid calli showed the highest production of TPC at 81.28 mg GAE/g extract, which was significantly (*P*-value = 0.00) higher than the diploid and mixoploid calli. The mixoploid calli had a TPC value followed by the tetraploid with a TPC value of 70.81 mg GAE/g extract. The diploid calli had the lowest TPC value of 35.59 mg GAE/g extract.

Total flavonoid content values obtained from the diploid, tetraploid, and mixoploid callus extracts were analyzed using the aluminium chloride method, with results shown in Table 2. The TFC showed the highest increase in the tetraploid calli with a TFC value of 35.33 mg QE/g extract, which was significantly (*P*-value = 0.00) higher than the diploid and mixoploid calli. The TFC value of the mixoploid extract was not significantly different (*P*-value = 0.00) from the diploid extract, with a TFC values of 18.84 and 18.67 mg QE/g

**Table 2 Dependent variables of phytochemical content and antioxidant activity of diploid, tetraploid and mixoploid *R. nasutus* calli.**

| Dependent variables | Ploidy levels | | | Standard |
|---|---|---|---|---|
| | Diploid | Tetraploid | Mixoploid | Ascorbic acid |
| Fresh weight (g) | $3.04 \pm 0.25^a$ | $3.45 \pm 0.33^a$ | $2.72 \pm 0.24^a$ | – |
| Dry weight (g) | $0.15 \pm 0.01^a$ | $0.14 \pm 0.01^{ab}$ | $0.11 \pm 0.02^b$ | – |
| Extraction yield (%) | $11.28 \pm 1.16^b$ | $27.56 \pm 1.31^a$ | $14.14 \pm 1.94^b$ | – |
| TPC (mg GAE/g extract) | $35.59 \pm 0.07^c$ | $81.28 \pm 0.21^a$ | $70.81 \pm 0.12^b$ | – |
| TFC (mg QE/g extract) | $18.67 \pm 0.33^b$ | $35.33 \pm 0.69^a$ | $18.84 \pm 1.07^b$ | – |
| FRAP assay (mg TE/g extract) | $12.65 \pm 0.23^c$ | $39.92 \pm 0.07^a$ | $28.81 \pm 0.25^b$ | – |
| $IC_{50}$ by DPPH assay ($\mu$g/mL) | $268.16 \pm 0.70^d$ | $89.13 \pm 0.20^b$ | $106.82 \pm 0.42^c$ | $5.40 \pm 0.01^a$ |
| $IC_{50}$ by ABTS assay ($\mu$g/mL) | $524.28 \pm 1.31^d$ | $179.17 \pm 0.32^b$ | $279.97 \pm 1.11^c$ | $3.18 \pm 0.00^a$ |

**Note:**
Mean ± SE values followed by different superscripts in the same row are significantly different according to ANOVA and Duncan's Multiple Range Test ($p < 0.05$).

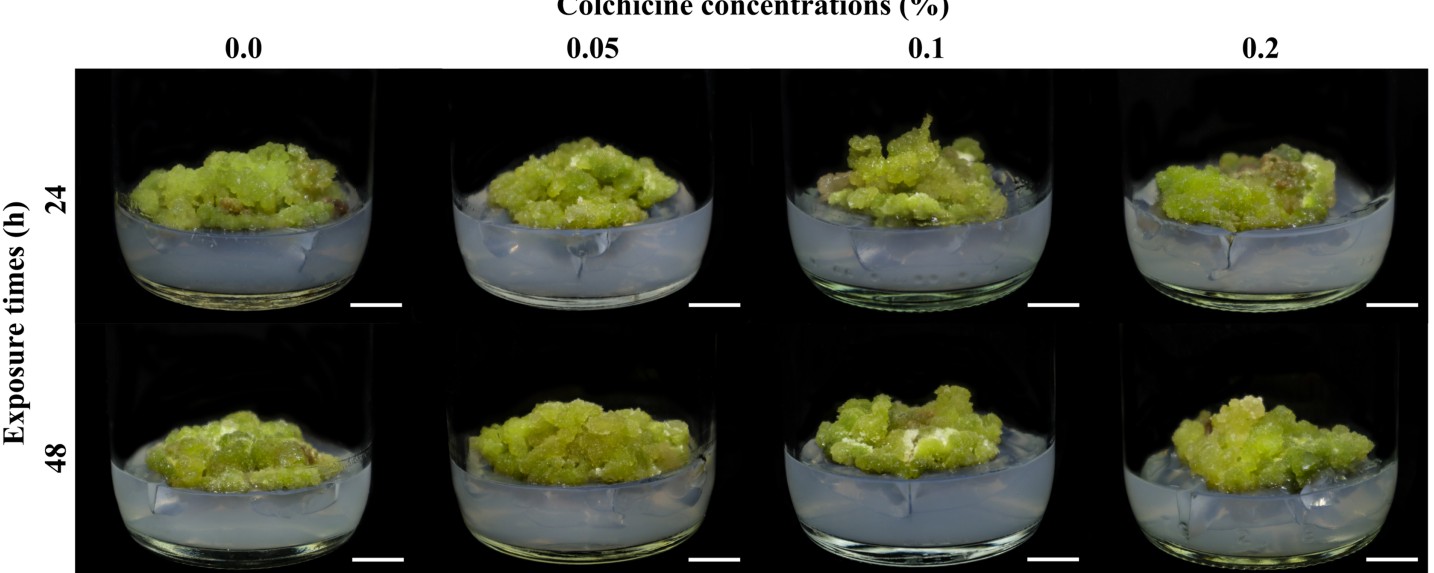

**Figure 2 Calli of *R. nasutus* after soaking in colchicine at different concentrations and exposure times and cultured on MS medium supplemented with 1 mg/L of kinetin plus 1 mg/L of 2,4-D for 4 weeks.** Scale bar = 1 cm.

extract, respectively. These results were consistent with the highest TPC production in callus extracts.

## Evaluation of antioxidant activities

The antioxidant activity values of *in vitro* polyploidy induction of diploid, tetraploid, and mixoploid calli were calculated using FRAP, DPPH, and ABTS assays. The FRAP reducing power scavenging activities are shown in Table 2. The FRAP assay showed the highest value increase in the tetraploid calli at 39.92 mg TE/g extract, which was significantly ($P$-value = 0.00) higher than the diploid and mixoploid calli. The FRAP values of the

methanolic extracts of mixoploid and diploid calli were 28.81 and 12.65 mg TE/g extract, respectively.

DPPH free radical scavenging activity of the diploid, tetraploid, and mixoploid calli were expressed as $IC_{50}$ values, as shown in Table 2. Higher antioxidant activity and more efficient DPPH scavenging are associated with lower $IC_{50}$ values. The standard ascorbic acid showed significantly (P-value = 0.00) highest activity by DPPH assay with an $IC_{50}$ value of 5.40 µg/mL. For polyploidy induction, the $IC_{50}$ value of the tetraploid callus extract showed significantly (P-value = 0.00) higher activity at 89.13 µg/mL compared to the mixoploid and diploid callus extracts. The $IC_{50}$ values by DPPH assay of mixoploid and diploid callus extracts were 108.82 and 268.16 µg/mL, respectively.

ABTS free radical scavenging abilities of the calli expressed as $IC_{50}$ values are shown in Table 2. The standard ascorbic acid showed significantly (P-value = 0.00) highest activity by ABTS assay with an $IC_{50}$ value of 3.18 µg/mL. Antioxidant activity evaluated in the tetraploid calli by ABTS gave an $IC_{50}$ value of 179.17 µg/mL, followed by the mixoploid callus extract with an $IC_{50}$ value of 279.97 µg/mL. The lowest ABTS radical scavenging was observed for the diploid callus extract with an $IC_{50}$ value of 524.28 µg/mL.

## Correlations and multivariate statistics

The correlation coefficients between phytochemical contents (TPC and TFC) with antioxidant scavenging activities (FRAP, DPPH, and ABTS assays) were studied using Pearson's correlation coefficients ($r$) at a 95% significance level ($p < 0.05$), as shown in Fig. 3. TPC showed significantly (P-value = 0.00) strong positive correlations with the FRAP assay ($r = 0.972$), and significantly strong negative correlations with the DPPH ($r = -0.992$) and ABTS assays ($r = -0.997$). By contrast, TFC showed a significantly strong positive correlation with the FRAP assay ($r = 0.731$), and a significantly weak negative correlation with the ABTS assay ($r = -0.625$), while a negative but not significant relationship ($p > 0.05$) was determined between TFC and the DPPH assay ($r = -0.465$). TFC also resulted in a significantly weak positive correlation with TPC ($r = 0.569$). The FRAP assay exhibited a significantly strong negative correlation with the DPPH ($r = -0.935$) and ABTS assays ($r = -0.986$). By contrast, the DPPH and ABTS assays showed a strong and significant positive correlation ($r = 0.646$).

PCA was applied to investigate the correlations between TPC and TFC and the FRAP, DPPH, and ABTS assays for the diploid, tetraploid, and mixoploid *R. nasutus* callus extracts. The biplot demonstrated that 99.9% of the total variance could be explained by two principal components. PC1 on the x-axis accounted for 87.2% of the total variability, while PC2 on the y-axis explained 12.7% of the total variability (Fig. 4). The symbols represent the three ploidy levels. The correlation of traits to PC1 and PC2 is indicated by the factor loading vectors (blue arrows). The loading vectors showed that TPC, FRAP, ABTS, and DPPH scavenging activities were strongly associated with PC1, while TFC was highly correlated with PC2. The correlation between variables is indicated by the angles between the vectors. In the PCA biplot, TPC had strong positive correlations with the FRAP assay and strong negative correlations with the ABTS and DPPH assays. The loading vector line length represents the variance level, with greater variance shown by

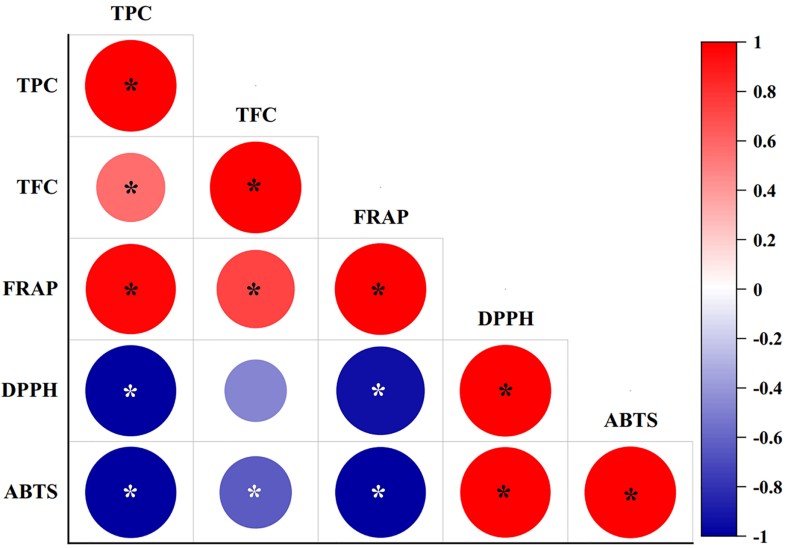

**Figure 3** Pearson's correlation coefficient matrix between phytochemical contents and antioxidant activities of diploid, tetraploid, and mixoploid calli extracts of *R. nasutus*. Significance level: *$p < 0.05$.

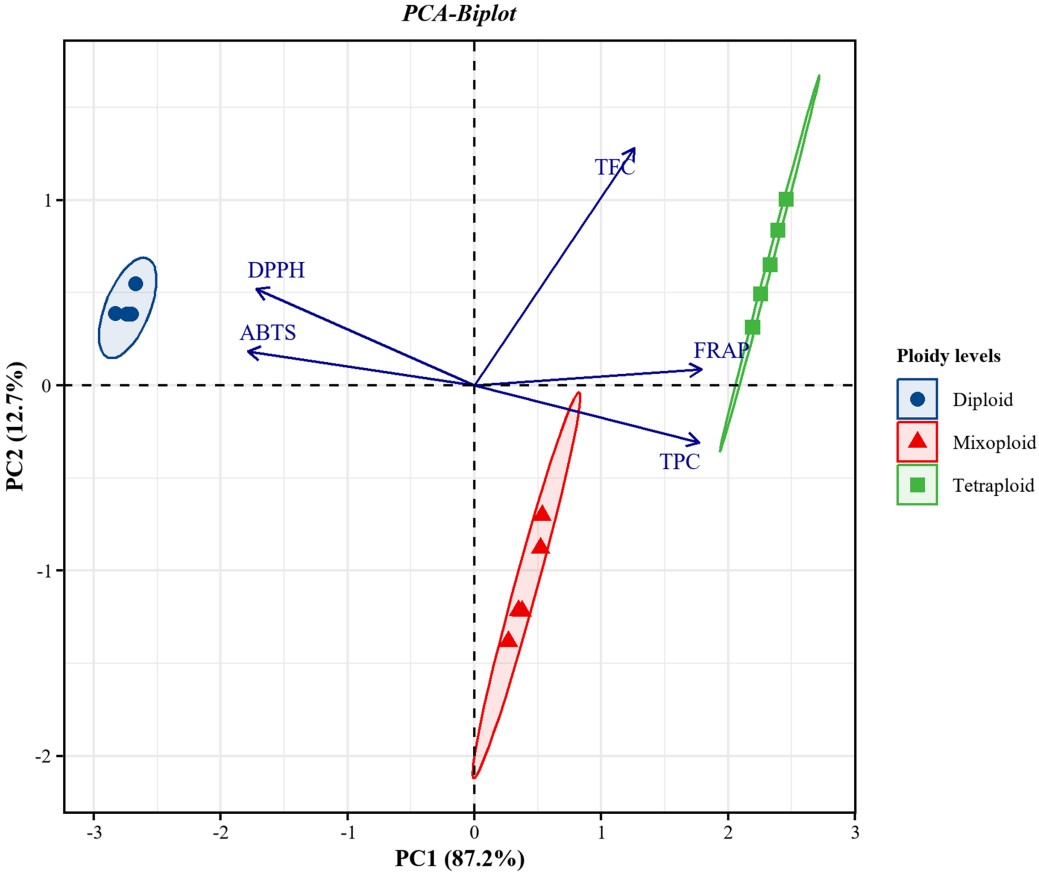

**Figure 4** Principal component analysis (PCA) biplot from PC1 and PC2 for diploid, tetraploid, and mixoploid calli extracts of *R. nusutus*, based on their phytochemical contents and antioxidant activities (blue arrows).

longer vector lines. The PCA biplot separated the diploid, tetraploid, and mixoploid calli in different quadrants. The distance between the score plot (circle, triangle, square symbols) and the loading line represented the value of the loading on the score plot. The tetraploid calli were closer to TPC, FRAP, and TFC than the DPPH and ABTS assays, indicating that tetraploids had higher TPC, FRAP, and TFC values. The PCA results were consistent with the experimental results mentioned above.

## DISCUSSION

### *In vitro* polyploidy induction

In our results, the highest rate of chromosome doubling (tetraploid calli) was accomplished at 0.05% colchicine for 48 h with a survival rate of 75%, while calli treated with 0.2% colchicine for 24 h were mixoploid with the lowest survival rate of 33.33%. Treated calli displayed signs of stress caused by colchicine. The severity of the symptoms and calli mortality varied according to colchicine concentration and exposure time. A similar result was found in a study on *Begonia* × *benariensis* (*Xie et al., 2024*). The most suitable treatment for polyploidy induction of calli was 0.05% colchicine for 4 h, with the highest survival rate at 46.67%. The survival rate of the treated calli varied from 18.33% to 41.67% at 0.1% colchicine concentration. Colchicine is toxic; thus, if the treatment duration is too long or the concentration is too high the plant materials die (*Manzoor et al., 2019*). The mortality rate of *R. nasutus* calli increased with higher colchicine concentrations and prolonged treatment durations. The toxic effects of elevated colchicine doses have also been documented in other studies focused on polyploid induction (*Wu et al., 2022*). The optimal combination of colchicine concentration and exposure time is necessary to ensure that the survival, growth, and development of the induced calli are not adversely impacted for the effective conversion of diploid plants into stable polyploidy.

The untreated diploid calli (0% colchicine) were used as the control, and the control and polyploidy calli were analyzed through flow cytometry. In this study, the fluorescence intensity in the histogram of diploid and tetraploid calli showed a peak value at channels of 200 and 400, respectively. The fluorescence intensity of tetraploid calli showed a peak at 400, 2-fold higher than the peak at 200, while mixoploid calli showed both channels 200 and 400 as equal. A similar result was found in a study on *Rhododendron fortune* (*Mo et al., 2020*). *The flo*w cytometry histogram showed that tetraploids and octoploids had a larger peak shift than uninduced diploid plantlets. Nuclear DNA quantification by flow cytometry is recommended as an alternative method for the screening of polyploidy plants.

Numerous plant species have successfully duplicated their chromosomes through the use of organogenesis mechanisms (*Eng & Ho, 2019*). Induced polyploidization has been used to produce a variety of morphological characteristics. Polyploid plants have larger cell sizes than diploid plants, known as the Gigas effect (*Sattler, Carvalho & Clarindo, 2016*). In our study, tetraploid calli showed higher fresh weight than diploid and mixoploid calli. A similar result was found in a study on somatic embryogenesis of *Panax vietnamensis* (*Diem et al., 2022*). Compared to diploids, tetraploids exhibited noticeably higher fresh weights

for both plantlets and rhizomes. Likewise, similar results were reported in *Melissa officinalis* (*Talei & Fotokian, 2020*), with tetraploids having higher fresh and dry weights than diploid plants. In a given plant species, the volume of the cell correlates with the quantity of DNA present; hence, as a plant transitions from diploid to tetraploid, the volume of the cell increases and the amount of DNA doubles (*Russell et al., 2004*; *Diem et al., 2022*). The observation that the dry weight of diploid calli was slightly more than tetraploid and mixoploid calli could arise due to several biological and physiological factors. While the Gigas effect, which involves increased cell size and biomass in polyploids, often predicts higher dry weight in polyploids, this is not always the case. Diploids may exhibit faster growth rates or more efficient metabolic processes than tetraploids, resulting in higher dry weight accumulation over the same period (*Pei et al., 2019*).

The effectiveness of an *in vitro* polyploidization procedure is influenced by various factors, including the availability of a well-established protocol for the *in vitro* regeneration of the target species, the type and concentration of the antimitotic agent, the duration of exposure, the method of application of the antimitotic solution, and the type of explant used (*Sattler, Carvalho & Clarindo, 2016*). However, the suitable conditions for inducing polyploidy could change depending on the differences in culture environment, culture medium, plant growth regulators, or plant genotype (*Niazian & Nalousi, 2020*), which might lead to a low rate of production of tetraploid cells and inevitably lead to mixoploid cells. In addition, callus is a collection of unorganized cells and after treatment with colchicine does not only result in a tetraploid cell. Therefore, a more precise procedure for inducing tetraploid cells in the *R. nasutus* calli is especially necessary. In future studies, the callus should undergo further purification following selection and regeneration to facilitate the induction of seedlings.

## Quantification of phytochemical and antioxidant activities

Artificial polyploidy induction has been demonstrated to enhance the synthesis of secondary metabolites in various species of medicinal plants compared to their diploid plants (*Gantait & Mukherjee, 2021*). In our study, tetraploid callus was observed to have a significantly higher percentage of extraction yield, TPC, and TFC values compared to diploid and mixoploid calli. A similar result was found in a study on *Solidago canadensis* (*Yang et al., 2021*), with tetraploid and hexaploid aerial plant parts having the highest amount of TPC throughout the vegetative, flowering, and fruiting stages. According to *Kong et al. (2017)*, tetraploid plants of *Lonicera japonica* had significantly higher amounts of TPC and TFC than diploid plants. Multiple sets of chromosomes produced from the same organism during polyploidization lead to the formation of autopolyploidy. The primary outcome of polyploidy is an increase in cell size brought about by the insertion of more gene copies. However, it is important to recognize that several epigenetic processes, such as DNA methylation, histone modification, and non-coding RNA activity, also play significant roles in regulating gene expression in polyploid plants (*Li & Chen, 2022*). Moreover, polyploidy plants with higher nuclear DNA content have higher gene

expression, which ultimately results in higher secondary metabolite synthesis. These metabolites are useful in pharmacology (*Manzoor et al., 2018*).

Phenolic compounds are mainly produced through the secondary metabolism of plants, and play a role in reducing reactive oxygen species and binding to lipid alkoxyl radicals to prevent lipid peroxidation (*Meitha, Pramesti & Suhandono, 2020*). They are classified as flavonoids, phenolic acids, simple phenols, and hydroxycinnamic acid derivatives (*Alara, Abdurahman & Ukaegbu, 2021*). The majority of phenolic compounds are flavonoids, with a basic structure of 15 carbon atoms arranged in two benzene rings connected by a heterocyclic pyrene ring (*Albuquerque et al., 2021*). Flavonoids possess antioxidant abilities and can decrease the risk of diabetes, cancer, Parkinson's, and Alzheimer's diseases (*Mark et al., 2019*). Polyploidization influences genetic composition and gene expression, enabling the development of novel regulatory pathways. This process enhances adaptability, broadens geographical distribution, and modifies community structures across diverse plant species (*Gupta et al., 2024a*). Gene duplication results in DNA amplification, which increases the copy number of specific genes. This amplification enhances mRNA expression, subsequently leading to the overproduction of key biosynthetic enzymes involved in the synthesis of secondary metabolites. Consequently, enzyme activity is elevated, and metabolic pathways are upregulated, thereby affecting the quantity, composition, and relative proportions of secondary metabolites (*Lavania et al., 2012*).

An antioxidant is a chemical substance that inhibits, or delays cell damage impacted by oxidants (*Bedlovičová et al., 2020*). Oxidative stress is induced by an excess of oxidants such as reactive oxygen species (ROS) or nitrogen species (RNS). The antioxidant activities of *R. nasutus* calli (diploid, tetraploid, and mixoploid) were evaluated and compared using the FRAP, DPPH, and ABTS assays. In our study, the methanolic extraction of tetraploid calli showed the highest antioxidant activity, as determined by the FRAP, DPPH, and ABTS assays, compared to diploid and mixoploid calli. A similar result was found in a study on *Thymus vulgaris*. The DPPH radical scavenging assay of tetraploid plants exhibited higher antioxidant potential with a lower $IC_{50}$ value than diploid plants (*Gupta et al., 2024b*). *Sanwal et al. (2010)* reported the antioxidant activity through the FRAP assay of the tetraploid extract of *Zingiber officinale* as higher than the diploid extract. These results supported the increase in antioxidant activities in polyploid plants compared to diploid plants.

The mechanisms of action of antioxidants include a hydrogen atom transfer mechanism or single-electron transfer *via* proton transfer, sequential proton loss electron transfer, and transition metal chelation (*Zeb, 2020*). In our study, TPC values were significantly correlated with the FRAP, DPPH, and ABTS assays. A similar result was reported by *Živković et al. (2019)*, with a significantly high correlation between the TPC value and the DPPH and ABTS assays. Antioxidant activity determination uses colorimetric assays that react with antioxidants. The FRAP assay measures the reduction of $Fe^{3+}$ (colorless) to $Fe^{2+}$ (deep blue) occurring in the presence of 2, 4, 6-trypyridyl-s-triazine. The most frequently used assays are the DPPH and ABTS assays (*Bedlovičová et al., 2020*). The DPPH assay is based on the reaction of DPPH$^{•}$ free radical (purple) to the reduced form DPPH

(hydrazine form, pale yellow) as a rapid, simple, and affordable assay (*Echegaray et al., 2021*). The ABTS assay is based on the reaction of the cation radical ABTS$^{\bullet+}$ (produced by oxidation, bluish green) to the reduced form ABTS (colorless) (*Cano et al., 2023*). The ABTS$^{\bullet+}$ radical can be used in aqueous or organic solvents (*Bedlovičová et al., 2020*). Results suggested that changes in the phytochemical contents and antioxidant activities were caused by polyploidization in *R. nasutus* calli.

## CONCLUSIONS

In conclusion, we found that the *in vitro* polyploidization of *R. nasutus* calli was successfully induced at 0.05% colchicine with 48 h of exposure time, giving tetraploid calli with higher fresh weight and percentage of extraction yield than diploid calli. The tetraploid calli showed significantly enhanced phenolic and flavonoid contents compared to the diploid calli. Moreover, tetraploid calli exhibited an approximately twofold higher increase in TPC and TFC compared to diploid calli. Antioxidant activity based on the FRAP, DPPH, and ABTS assays indicated that the tetraploid calli had higher ability than the diploid calli. Pearson's correlation and the PCA results indicated correlations between phytochemical contents and antioxidant activity assays. The FRAP, DPPH, and ABTS assays also showed strong correlations with the phenolic content. However, to obtain truly tetraploid cell lines in callus culture, it is necessary to establish calli after selection and evaluation of ploidy level. Polyploidization was determined as a potential method to induce tetraploid *R. nasutus* calli and enhance phenolic and flavonoid contents and antioxidant activity for medical purposes.

### Funding

This study was financially supported by the Science Achievement Scholarship of Thailand (SAST). The funders had no role in study design, data collection and analysis, decision to publish, or preparation of the manuscript.

### Grant Disclosures

The following grant information was disclosed by the authors:
Science Achievement Scholarship of Thailand (SAST).

### Competing Interests

The authors declare that they have no competing interests.

### Author Contributions

- Wipa Yaowachai conceived and designed the experiments, performed the experiments, analyzed the data, prepared figures and/or tables, authored or reviewed drafts of the article, and approved the final draft.
- Prathan Luecha analyzed the data, authored or reviewed drafts of the article, and approved the final draft.
- Worasitikulya Taratima conceived and designed the experiments, analyzed the data, authored or reviewed drafts of the article, and approved the final draft.

## DNA Deposition

The following information was supplied regarding the deposition of DNA sequences:

The flow cytometry is available at figshare: Yaowachai, Wipa (2025). FCS files (polyploidy of *R. nasutus* callus). figshare. Dataset. https://doi.org/10.6084/m9.figshare.28494230.v1.

## Data Availability

The raw measurements are available in the Supplemental File.

## Supplemental Information

Supplemental information for this article can be found online at http://dx.doi.org/10.7717/peerj.19160#supplemental-information.

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
