# Peer review of "Phytochemical production and antioxidant activity improvement of Rhinacanthus nasutus (L.) Kurz calli by in vitro polyploidization"

_PeerJ, doi:10.7717/peerj.19160_

## Round 0.1 · original submission · Minor Revisions

Your manuscript was reviewed by four independent experts in the field. All four reviewers find the work interesting but raised several issues which need to be addressed properly. The reviewers provide comments in their reviews and pointed out the areas where the manuscript needs to be improved. I also read the manuscript carefully and largely agree with the reviewers’ comments.

Reviewer 1 ·

Basic reporting

There is need to review the genetic composition and karyotype in the species -Rhinacanthus nasutus before engaging in tetrapolyplodization.
Demonstrating why colchicine was used in the study instead of trifluralin or oryzalin will improve the quality of the paper.
line 310-317- that section will be best placed in the introduction to avoid pure literature review in the discussion.
lines 352 and 353- This statement may not fit well in the discussion, read like and objective or aim statement, it needs to be rephrased.
lines 359-361- Rephrase the statement for clarity.

Experimental design

line 93 to line 102- Clearly explain the experimental design utilized in this invitro activities
line 194- How was CRD applied in the antioxidant assays?

Validity of the findings

Table 2: Why does the diploid possess higher dry-weight than the tetraploid and the mixoploid - would this contravene the Gigas effect.
Table 2: line 4; Correct the notes section to read rows instead of roll.
Qualify statement in line 373-375, noting that several epigenetic processes exist.

Additional comments

Improve the abstract by briefly quantifying the result statements.

Reviewer 2 ·

Basic reporting

NO COMMENTS

Experimental design

ALL EXPERIMETS HAVE BEEN DESIGNED AND EXPLAINED ADEQUATELY IN THE MANUSCRIPT

Validity of the findings

NO COMMENTS

Additional comments

THE MANUSCRIPT IS OF A SIGNIFICANT CONTRIBUTION IN THE SAID FIELD.

·

Basic reporting

Article is written in clear english and found to be technically correct.

References additions should be enhanced in order to support and authenticate the findin

Experimental design

Section Materials and Methods:

There is not clear methodology explained for sterilization techniques. author should explain that which surfactant (and conectarations) was used for sterilization technique .
author mentioned that The calli were soaked in a filter-sterilized solution of colchicine at various concentrations (0, 0.05, 0.1, and 0.2 %) for 24 and 48 h. It should be explained that how these concentration were selected, if selected from old literature then it should have been mentioned .

Likewise there is no mention of any reference for the flow cytometry analysis. all the methods should be labeled with the references from where they have been adopted,

similarly there is also need to cite the figures where there is mention of figures

Validity of the findings

Results are supported by robust data.

Discussion should be more elaborated with logical comments and relevant references

Additional comments

No comments

·

Basic reporting

Please check attachment file.

Experimental design

Please check attachment file.

Validity of the findings

Please check attachment file.

Additional comments

no comment

---

## Round 0.2 · Minor Revisions

Authors have satisfactorily addressed most of the comments. However, the Section Editor has recommended the following revision to the abstract:

"Please avoid acronyms in the abstract that are not explained (FRAP, DPPH, and ABTS assays). Either letter-out the acronyms, or just write 'antioxidant activity.' Also state in the Abstract and Conclusions the increase of bioactive compounds; e.g. by x%."

Reviewer 1 ·

Basic reporting

OK

Experimental design

OK

Validity of the findings

OK

Additional comments

NON

---

## Round 0.3 · accepted · Accept

The current version is satisfactory and is ready for publication.

The Section Editor has requested that the authors present their sample sizes either in the Methods or in the data table(s). This can be addressed during the Proof stage.